# Acoustic Wave-Driven Liquid Metal Expansion

**DOI:** 10.3390/mi13050685

**Published:** 2022-04-28

**Authors:** Youngbin Hyun, Jeong-Bong Lee, Sangkug Chung, Daeyoung Kim

**Affiliations:** 1Department of Mechanical Engineering, Myongji University, Yongin 120-728, Korea; ieeh0450@gmail.com; 2Department of Electrical Engineering, The University of Texas at Dallas, Richardson, TX 75080, USA; jblee@utdallas.edu; 3Department of Electrical Engineering, Korea Army Academy at Yeongcheon, Yeongcheon 770-849, Korea

**Keywords:** liquid metal droplet, acoustic wave, expansion, oxidation, oxide crack

## Abstract

In this paper, we report a volume expansion phenomenon of a liquid metal droplet naturally oxidized in an ambient environment by applying an acoustic wave. An oxidized gallium-based liquid metal droplet was placed on a paper towel, and a piezo-actuator was attached underneath it. When a liquid metal droplet was excited by acoustic wave, the volume of liquid metal was expanded due to the inflow of air throughout the oxide crack. The liquid metal without the oxide layer cannot be expanded with an applied acoustic wave. To confirm the effect of the expansion of the oxidized liquid metal droplet, we measured an expansion ratio, which was calculated by comparing the expanded size in the *x* (horizontal), *y* (vertical) axis to the initial size of the liquid metal droplet, using a high-speed camera. For various volumes of the droplet, when we applied various voltages in the range of 5~8 Vrms with 18.5~24.5 kHz using the piezo-actuator, we obtained a maximum expansion ratio of 2.4 in the *x* axis and 3.8 in the *y* axis, respectively. In addition, we investigated that the time to reach the maximum expansion in proportion to the volume size of liquid metal differed by five times from 4 s to 20 s, and that the time to maintain the maximum expansion differed from 23 s to 2.5 s, which was inversely proportional to the volume size. We also investigated the expansion ratios depending on the exposure time to the atmosphere. Finally, a circuit containing LED, which can be turned on by expanded liquid metal droplet, was demonstrated.

## 1. Introduction

The gallium-based liquid metal alloy has both metallic properties such as high thermal and electrical conductivities and liquid properties such as infinite shape deformation [1,2]. In addition, gallium-based liquid metal maintains a liquid phase at room temperature and is not toxic unlike mercury [3]. Due to these properties, liquid metal alloys have been studied in many applications such as soft robotics [4], motor [5], switches [6,7], direct-write 3D printing [8,9], tunable antennas [10,11], metamaterials [12,13,14], and flexible and wearable devices [15,16]. In many applications, it is highly desirable to have the capability of controlling the position of liquid metal alloy on demand. However, liquid metal is instantly oxidized in ambient air with a self-terminating thin oxide layer of 0.5–3 nm [17], which allows the liquid metal droplet to attach itself to most surfaces [18]. Many studies have been conducted to mobilize liquid metal on engineered surfaces [19,20], using various forces such as electrical [21], magnetic [22], and acoustic [23] forces. Additionally, the deformation of liquid metal has been widely investigated. For example, H. Wang et al. conducted the deformation of a mixture of silicon rubber and liquid metal by applying heat [15]. J. Jeon et al. conducted the shape control of magnetic liquid metal marble (liquid metal coated with an iron powder) by applying magnetic force by chemical reaction [24]. These studies controlled the shape of liquid metal by utilizing external energy to control the mixed substances rather than controlling pure liquid metal. There was an experiment to transform liquid metal itself by using the laser without direct contact. M. S. Krivokorytov et al. and S. Y. Grigoryev et al. applied a laser pulse to deform liquid metal in a vacuum, creating cavitation inside liquid metal [18,25].

In order to utilize liquid metal more efficiently, the size control of liquid metal has also been studied as micro-/nano-scale liquid metal droplets can be widely applied to various applications, including soft electronics [26,27]. However, controlling the size of liquid metal droplets requires a particular condition due to the naturally formed oxide layer and strong surface tension of liquid metal. As one of the methods studied to reduce the effect of the oxide layer, micro-/nano-scale liquid metal droplets were controlled using an acoustic wave. For instance, S.-Y. Tang et al. conducted a study using sonication to separate a liquid metal droplet into fine droplets [28], and A. Yamaguchi et al. sought to control liquid metal on a nanometer scale using ultrasonication [29]. T. Yang et al. demonstrated a thermal switch using the millimeter scale-liquid metal droplet [30]. However, in order to apply liquid metal to a thermal switch, it required a sophisticated channel and needed the additional actuator that could move liquid metal.

This paper proposes a method to control the size of oxidized liquid metal using an acoustic wave in ambient atmosphere. We investigated the acoustic wave-driven volume expansion of an oxidized liquid metal droplet maintaining electrical conductivity. However, without the oxide layer on the liquid metal surface, atomization (the separation of small droplets) was observed. For liquid metal droplets with an oxide layer, we studied the effect of frequencies/voltages on the volume expansion and times to reach to the maximum expansion, and the times to maintain the maximum expansion. The exposure time to air environment (oxidation time)-dependent expansion was also investigated. An experiment showing the electrical conduction of the expanded liquid metal was finally demonstrated. 

## 2. Mechanism and Experiment Setup

A schematic diagram of the experimental setup for the study of liquid metal droplet expansion by acoustic wave is illustrated in Figure 1. Figure 1a shows the schematic diagram of the expansion principle of liquid metal droplets with an applied acoustic wave. A paper towel was utilized as a substrate as it was not wettable to the oxidized Galinstan droplet [18]. We placed a ~25 μL liquid metal droplet onto the paper towel. We chose the paper towel as a substrate as it is super-lyophobic to an oxidized liquid metal droplet [19]. The effect of the paper towel as a substrate is that it can change the resonant frequency of the expanded liquid metal droplet. As the acoustic wave was applied to the liquid metal throughout the paper towel, the substrate changed the resonant frequency for different volumes of droplets. Underneath the paper towel, a piezo-actuator (KPR3020LC-450, Daeyoung Electric Co., Seoul, Korea) was attached and the acoustic wave was applied by controlling the frequency and voltage amplitude of the signal. The signal was generated with a function generator (33210A, Agilent Co., Santa Clara, CA, USA) and an amplifier (PZD700, Trek Co., Denver, CO, USA). When the generated signal was applied to the piezo-actuator, the acoustic wave was produced, and it was delivered to the droplet through the paper substrate. When the voltage above 3 Vrms and above 1 kHz was applied to the piezo-actuator, the liquid metal droplet was rapidly oscillated. An oxidized liquid metal droplet consists of pure liquid metal inside and a ~3 nm native solid gallium oxide (Ga_2_O or Ga_2_O_3_) layer at the outmost surface [31,32]. The solid oxide layer on an oxidized liquid metal droplet oscillates according to the externally applied vibrational energy, which develops cracks on the solid surface oxide layer. The ~3 nm oxide layer ruptures as cracks are developed; however, exposed liquid metal inside is instantaneously oxidized upon exposure to air and forms a new oxide layer [33]. This crack is a space where the air flows into the droplet. Thus, liquid metal was expanded due to the inflowing air. When the frequency of the applied acoustic wave was matched with the resonance frequency of the droplet, the oscillation and expansion of the droplet is maximized. The resonance frequency of liquid metal is dependent on the volume of the liquid metal droplet. Figure 1b shows the optical images of an expanded oxidized liquid metal droplet with an applied acoustic wave (please refer to the Appendix A). When the acoustic wave was applied, the surface started to oscillate and gradually expanded, as shown in Figure 1(b2). Figure 1(b3) is an optical image of maximally expanded liquid metal.

When oxidized liquid metal was treated with HCl, the oxidized surface layer was removed [34]. After the HCl treatment, Figure 1(c1) shows a more spherical shaped-liquid metal droplet, compared to the oxidized liquid metal droplet (Figure 1(b1)). By applying an acoustic wave to a liquid metal droplet without the oxide layer, the droplet was oscillated, as shown in Figure 1(c2). When the voltage was increased and reached 7 Vrms or higher, liquid metal droplet was atomized. Figure 1d is a schematic of the experimental setup used to carry out the study. All experimental results were recorded using high-speed cameras (Phantom Miro X4, Vision Research, Inc., Wayne, NJ, USA) and CCD (Charge-coupled device) camera (EO-1312C, Edmund Optics). The deformability was measured by comparing the expanded size in the *x*, *y* axis to the initial size of the liquid metal droplet using a high-speed camera. The deformability was experimentally investigated depending on various applied frequencies and voltages, volumes, and oxidation times, which means the exposed time of liquid metal to the ambient environment before experiment execution.

In order to confirm the oxide crack, we captured the optical and SEM images of expanded liquid metal, as shown in Figure 2. Figure 2a shows the 20 μL liquid metal droplet before the expansion. As shown in Figure 2b, the liquid metal droplet was expanded when the acoustic wave was applied. The surface of the expanded liquid metal droplet was coated with a rough oxide layer, as shown in Figure 2c. Figure 2d shows the SEM image of an oxide layer of expanded liquid metal and a red-colored circle dot, indicating that the crack (~5 μm) was created by expanding the liquid metal droplet on the oxide layer. Therefore, we could visually confirm the oxide layer crack, whose path of air inflow was causing the expansion of liquid metal. 

## 3. Results and Discussion

In order to analyze the size change of the liquid metal droplet in the *x* and *y*-axis directions with/without an applied acoustic wave, we investigated the frequency dependent deformability of the expanded liquid metal. Figure 3 shows the deformability in the *x* (horizontal) and *y*-axis (vertical) directions of the expanded liquid metal droplet as a function of applied frequency. In order to verify the size changes of the liquid metal droplet, various volumes of liquid metal droplet (10, 15, 20, and 25 μL) were used, and we applied the acoustic wave using the piezo-actuator, whose frequencies are from 1.85 kHz to 2.45 kHz, with intervals of 0.05 kHz at 6 Vrms. 

The expansion phenomena depending on diverse volumes of liquid metal droplets occurred only in a certain range of frequencies. We observed that expansion phenomena did not occur with a frequency lower than 1.85 kHz and greater than 2.45 kHz for various volumes of liquid metal droplet. For example, for the 25 μL liquid metal droplet with an applied acoustic wave larger than 2.05 kHz and lower than 1.85 kHz frequency, only surface oscillation was observed without expansion. We defined resonant frequencies as frequencies that occur when the maximum volume expansion of the liquid metal droplet is obtained. Based on Figure 3, the resonant frequencies are 2.3, 2.2, 2, and 1.9 kHz, respectively, for volumes of 10, 15, 20, and 25 μL liquid metal droplets. Those liquid metal droplets were placed in air environment for 4 h (oxidized in 4 h).

There was a tendency that the changes of the deformability in the *x*-axis direction were similar to those in the *y*-axis direction and the resonant frequencies of the liquid metal droplets increased as the liquid metal droplet volume decreased. The maximum deformability in both the *x* (2.4 times) and *y*-axis (3.8 times) was obtained with a 20 μL liquid metal droplet. We expected that the paper towel underneath the droplet oscillated with the applied acoustic wave. Thus, it also has a resonant frequency where the paper towel can optimally follow the oscillation, and it can change the expansion ratio of various volume’s droplet. Therefore, the maximum volume expansion occurred with the 20 μL liquid metal droplet. 

The deformability of the liquid metal droplet depending on the voltages applied by the piezo-actuator was shown in Figure 4 in order to verify the impact of the applied voltages on the expansion phenomenon. 

We measured the deformability of liquid metal droplets in the maximum expansion state, with volumes of 10, 15, 20, and 20 μL, with the applied voltages between 5 Vrms and 8 Vrms, and with the interval of 0.5 Vrms at resonant frequencies for various droplet volumes. From the quantification results from the graphs, we confirmed the tendency that the deformability in the both the *x* and *y*-axis directions of the liquid metal droplet against the applied voltages increased, except for the liquid metal droplet with the volume of 20 μL, which showed a different trend compared to the other three liquid metal droplets with diverse volumes. We expected that 6 Vrms and 2 kHz make up the optimum condition for a 20 μL liquid metal droplet. Again, this was affected by the paper towel underneath the droplet. In addition, the changes of the deformability in the *x*-axis direction were similar to those in the *y*-axis direction as the applied voltages increased. We verified that the voltages applied to the piezo-actuator contributed to the *x* and *y*-axis extension of the volume of the expanded liquid metal droplet, except for the liquid metal droplet with the volume of 20 μL.

We also investigate the time needed to reach the maximum expansion, and the maintenance time to keep the maximum expansion, which depends on the various volumes of liquid metal droplets, as shown in Figure 5. The experiment was performed by applying the same voltage of 6 Vrms and the resonance frequencies for 10, 15, 20, and 25 μL liquid metal droplets, respectively. The 25 μL liquid metal droplet took the longest time to reach maximum expansion, with an average of 21.3 s, and after reaching it, and the maximum expansion was maintained with an average of 7 s. On the other hand, the 10 μL liquid metal droplet took the shortest time to reach maximum expansion, with an average of 3.9 s, and after reaching it, it was observed that the maximum expansion was maintained for a long time compared to the other three volume’s droplets, with an average of 91.5 s. According to this experiment, we confirmed that the required time to reach the maximum expansion increased as the volume of the liquid metal droplet increased. However, the maintaining time to keep the maximum expansion decreased as the volume of the liquid metal droplet increased. We expected that when the vibration energy was applied to the liquid metal continuously, the crack on the oxidized surface was recreated and recovered due to re-oxidation; thus it could keep the droplet expanded for a certain time. However, after maintaining the time to keep the maximum expansion, the inflowed air could not stay inserted in the droplet due to the oxide crack recovery on the surface resulting from the re-oxidation. 

Additionally, we found that when the applied acoustic wave was removed, the expanded volume liquid metal was decreased but could recover its original volume. After the maximum expansion of the liquid metal droplet, which was 2.3 times larger than the initial state in the *y*-axis direction, we investigated the recovery by removing the acoustic wave. The maximum expansion of the 25 μL liquid metal droplet was obtained with an applied acoustic wave (1.9 kHz and 6 Vrms), and we removed the acoustic wave. As shown in Figure 6, the volume of expanded liquid metal droplet was quickly reduced for 10 min because the inflowed air escaped back into the atmosphere through the oxide crack. Even after 60 min recovery, droplet could not be recovered to the original size, and it was 1.6 times larger than the initial state, which is the recovery limit. Then, we applied the acoustic wave again to the recovered liquid metal droplet to confirm the possibility of re-expansion. The liquid metal droplet expanded again; its expansion was smaller than the first maximum expansion but twice as large as the initial state. When we removed the acoustic wave, we observed the same trend as the first recovery phenomena, which is that it was quickly reduced for 10 min and gradually recovered for 60 min, resulting in saturation at 1.7 times expansion. When the repeated experiments were executed, the apparent degradation in deformation was observed. We expected that the thickness of the oxidation layer was increased as the consecutive acoustic wave had been applied, resulting in the size reduction of the cavity on the surface. As shown in the Figure 6 insets, a more roughened oxidized surface was observed. Therefore, the expansion ratio kept decreasing with consecutive experiments.

In addition, we investigated the expansion phenomenon depending on the oxidation time when liquid metal droplet (~25μL) was exposed to the ambient atmosphere in order to verify the impact of the oxide layer on the expansion phenomenon, as shown in Figure 7. In all experiments, the acoustic wave at 2 kHz with the voltages of 6 Vrms from the piezo-actuator was applied to liquid metal droplet. The deformability in the *x*-axis (red-colored bar graph) and *y*-axis directions (black-colored bar graph), according to the oxidation time with the interval of one hour from 0 to four hour and 24 h between 24 and 48 h, was analyzed. We calculated the deformability by dividing the expanded sizes in *x* and *y*-axis directions into their initial sizes. From the quantification results shown in Figure 7, as the exposed time for liquid metal droplets to the ambient atmosphere increased, both the deformability in the *x* and *y*-axis directions of the liquid metal droplet increased. The liquid metal droplet was more deformed in the *x*-axis direction than in the *y*-axis direction, for all sections (0~48 h). After the exposed time of 4 h, the deformability in both the x and *y*-axis directions was saturated, without considerable differences. From 0 to 0.5 h, the liquid metal droplet was deformed, with the greatest changes occurring in the both the *x* and *y*-axis directions (16% and 40%, respectively). We verified that the oxide time on the liquid metal surface had an influence on the expansion phenomenon of the liquid metal droplet and extended the expanded volume. This is a controversial result as the surface oxidation occurred instantly, and its thickness was ~3 nm, even over time, in the absence of physical perturbation [35,36]. Thus, we expected that the oxidation thickness was not much changed, but the chemical strength between the oxides is changeable. Thus, the expansion phenomenon depending oxidation time changed. We found that the deformability was almost saturated with 4 h oxidation in an ambient environment. We also expect that the expansion phenomenon could be affected by temperature as it is heavily related to the oxidation. 

In order to demonstrate the electrical conductivity when the liquid metal droplet expanded, we experimentally demonstrated, as shown in Figure 8, that the illumination of the LED using the expanded liquid metal droplet worked as an electrical switch (please refer to Appendix A). Figure 8a shows the conceptual schematic of the experimental circuit that was connected by the expansion of the liquid metal droplet in the middle part of it. We drew the silver paste line on the paper towel for the connection between the liquid metal droplet and the DC supply in order to minimize the contact between the liquid metal droplet and the wire. Figure 8b shows an actual experimental image, and we applied the voltage of 1.9 kHz and 8 Vrms to a piezo-actuator placed on a 25 μL liquid metal droplet to generate an acoustic wave for 2 s. When the liquid metal droplet with the acoustic wave was expanded, the LED was turned on due to connecting between the liquid metal droplet and the wire. According to this experiment, we confirmed that liquid metal droplet maintains electrical conductivity even when expansion occurs.

## 4. Conclusions

In this paper, we investigated the volume expansion phenomenon of the oxidized liquid metal droplet with an applied acoustic wave throughout the paper towel. When the acoustic wave at 2 kHz and 6 Vrms was applied to the 20 μL droplet, a maximum expansion ratio of 2.4 in the *x* axis and 3.8 in the *y* axis was obtained, respectively. In addition, the acoustic wave-applied time needed to reach the maximum expansion, and the time needed to maintain the maximum expansion, varied in relation to the volumes of the droplets. After 4 h of oxidation in an air environment, the deformability of the maximum expansion size of the liquid metal droplet was saturated. Finally, the maintenance of the electric conductivity for an expanded liquid metal droplet was successfully demonstrated by turning on the LED. We expected that acoustic wave-driven volume expansion would contribute to the electrical switching, triggering, and frequency-tuning of RF applications using oxidized liquid metal as long as the deforming speed could be accelerated.

## Figures and Tables

**Figure 1 micromachines-13-00685-f001:**
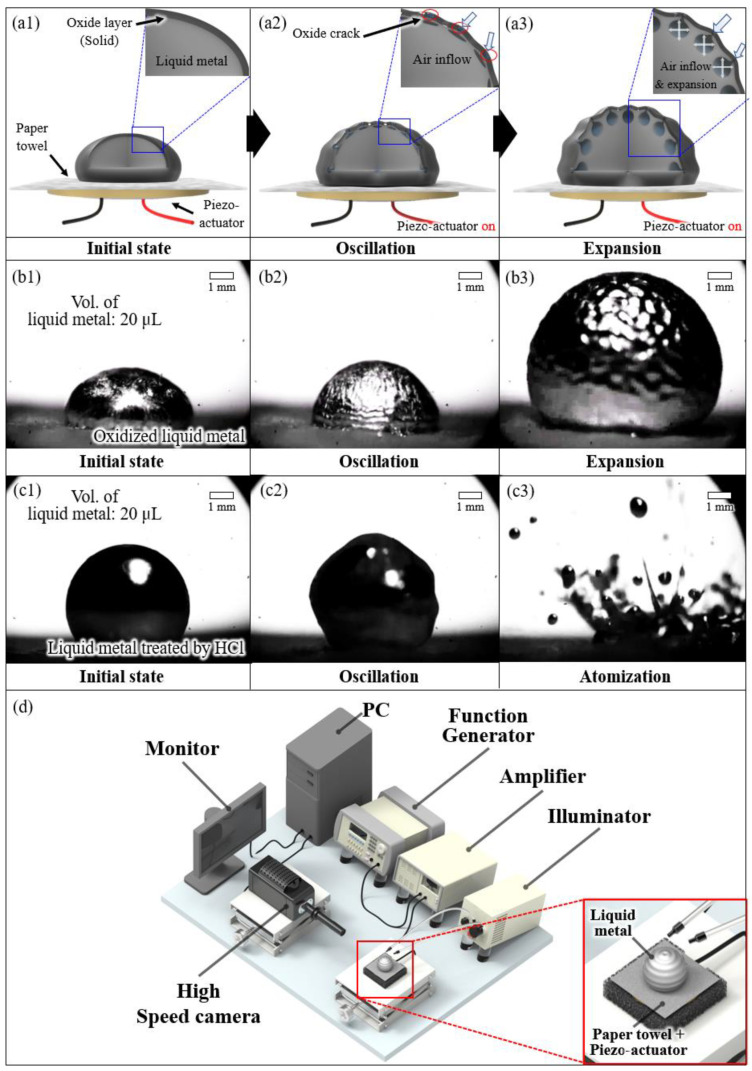
Liquid metal droplet expansion by acoustic wave: (**a**) the conceptual schematic of the liquid metal droplet expansion mechanism, (**b**) captured images of an expanded liquid metal droplet, (**c**) captured images of separated droplets for liquid metal treated with HCl, and (**d**) a schematic diagram of experimental setup.

**Figure 2 micromachines-13-00685-f002:**
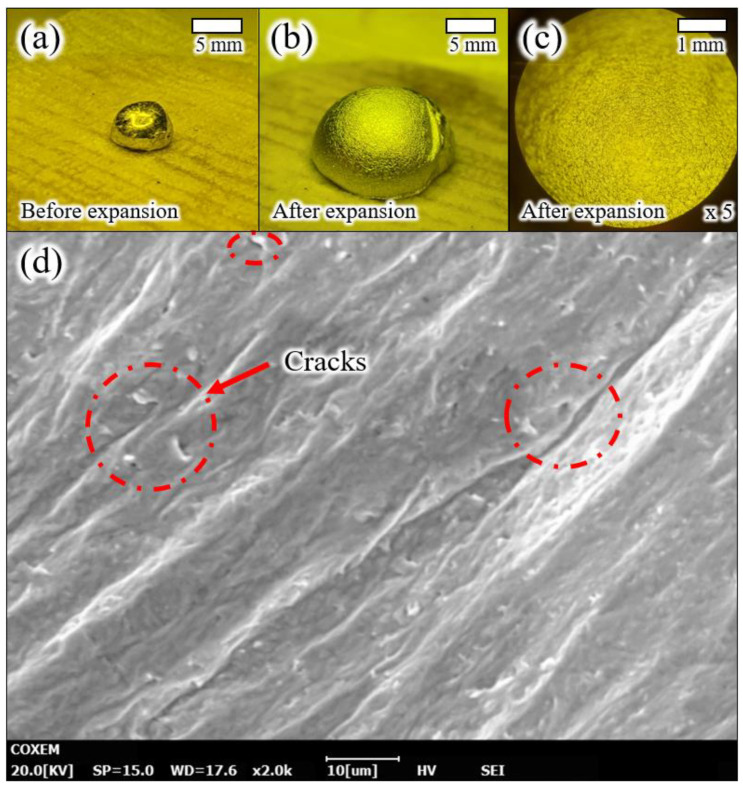
(**a**) An optical image of liquid metal droplets before expansion, (**b**,**c**) optical images of an expanded liquid metal droplet, and (**d**) SEM images of oxide cracks on an expanded liquid metal surface.

**Figure 3 micromachines-13-00685-f003:**
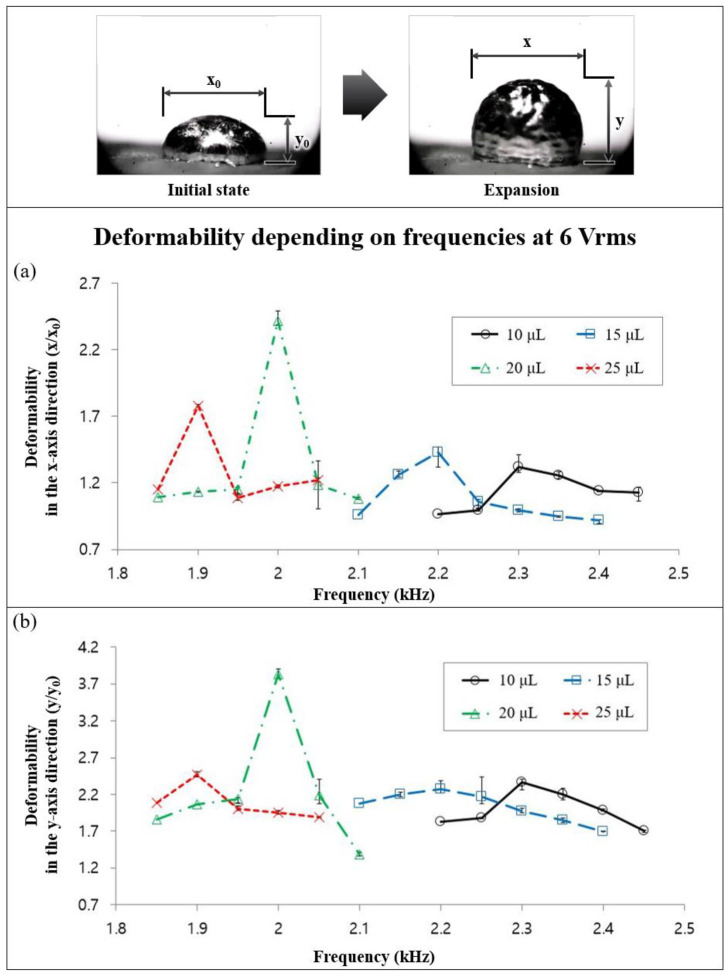
The deformability of liquid metal droplets in the (**a**) *x* and (**b**) *y*-axis direction, depending on applied frequencies from the amplifier.

**Figure 4 micromachines-13-00685-f004:**
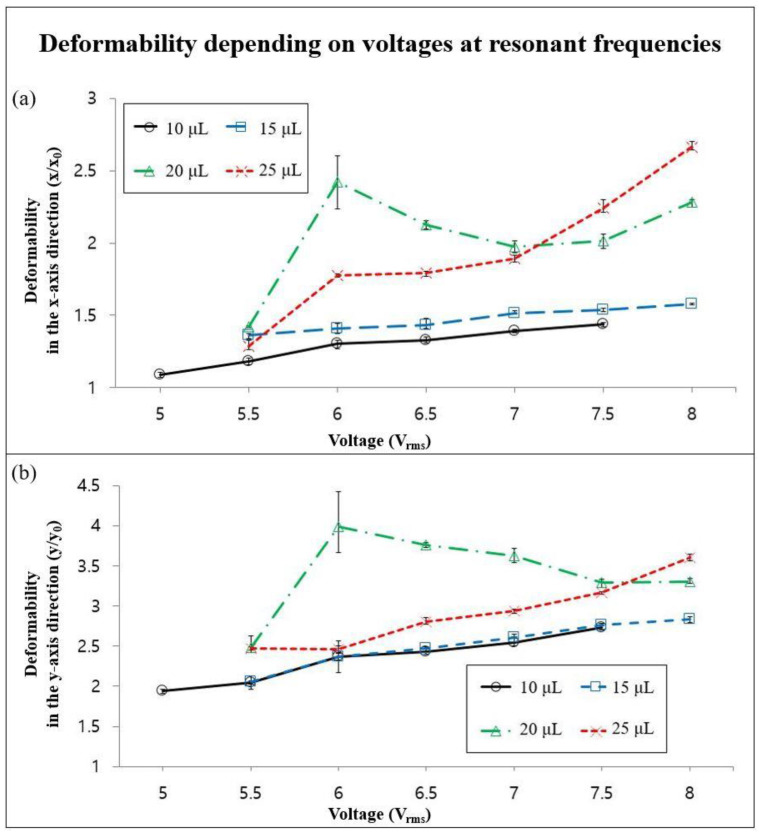
The deformability of liquid metal droplets in the (**a**) *x* and (**b**) *y*-axis direction, depending on the voltage of the acoustic wave for the piezo-actuator.

**Figure 5 micromachines-13-00685-f005:**
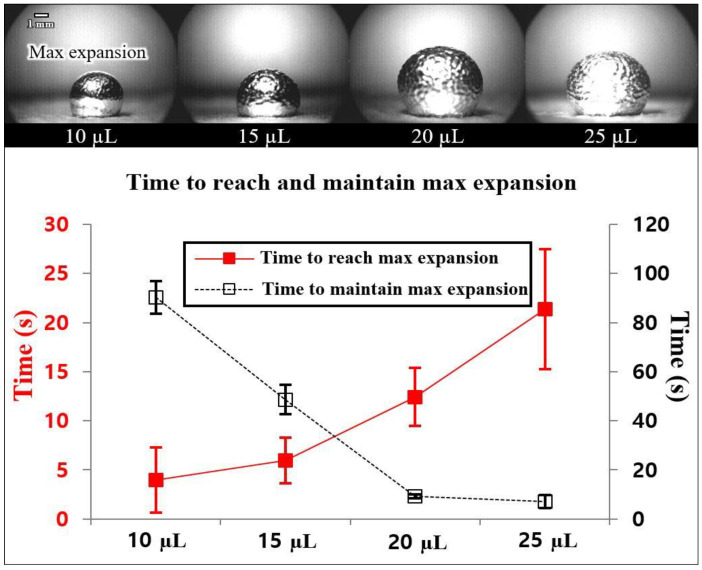
The time needed to reach and maintain maximum expansion, depending on the size of the liquid metal droplet.

**Figure 6 micromachines-13-00685-f006:**
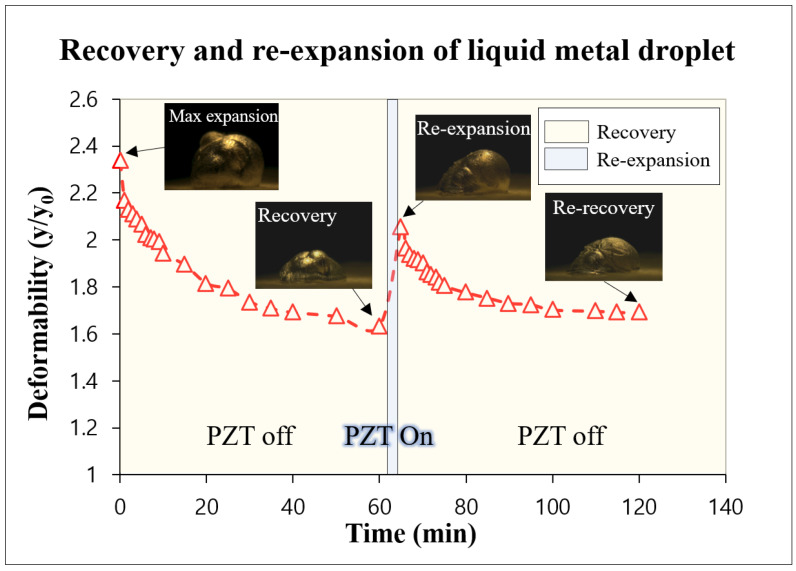
The deformation of the liquid metal droplet in the *y*-axis direction with/without an acoustic wave after maximum expansion.

**Figure 7 micromachines-13-00685-f007:**
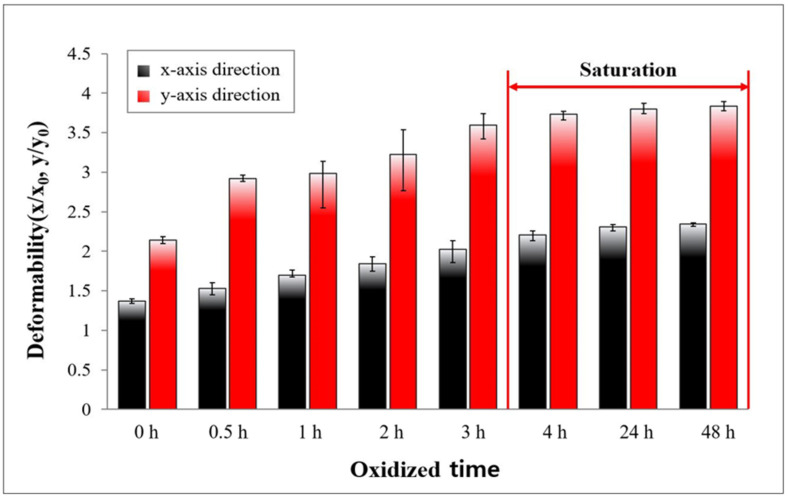
The deformation of liquid metal droplets in the *x* and *y*-axis direction, according to oxidation time.

**Figure 8 micromachines-13-00685-f008:**
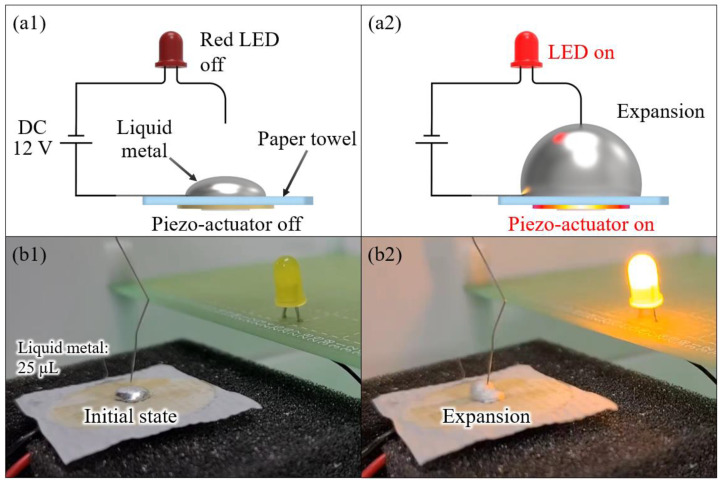
(**a**) A conceptual schematic of the circuit using liquid metal droplet expansion for an electrical switching application; and (**b**) the time-lapse images of the maintenance of electric conductivity on the expansion of a liquid metal droplet.

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
