# Peer review of "Acoustic Wave-Driven Liquid Metal Expansion"

_micromachines, 2022, doi:10.3390/mi13050685_

Round 1

Reviewer 1 Report

The authors report on an interesting physical/chemical effect, the controlled expansion of a liquid metal droplet by application of acoustic waves.

The paper presents a comprehensive study of the performed experiments, discusses the potential physical and chemical mechanisms behind, and closes with a simple demonstration of application.

Despite the fact, that the presented study represents thorough and comprehensive work, certainly of interest for the scientific community, the manuscript needs some fundamental improvements.

  • The examined effect of volume expansion obviously degrades when repeated (as clearly visible, for example, in Fig. 6). The authors should comprehensively comment on issues like repeatability and degradation. Further experiments concerning long term operation or at least showing multiple operation cycles should be included in the study.
  • The effect of the substrate underneath, the paper towel, is not well analyzed and described.
  • The presented explanation for the limited time to hold the max expansion is not well elaborated and described. Especially the section between line 190 and 195 is a bit confusing for the reader. In general, the supposed oxidation mechanism behind the expansion effect with air cavities underneath the surface, as illustrated in Fig. 1a, is just an assumption and not supported by any model/simulation. Can the authors provide experimental evidence or supporting simulations/calculations?
  • The “Colclusion” is rather a summary than a real conclusion, the authors should elaborate and point out potential applications in more detail. The effect seems to be rather slow and degrades (most probably) fast, with unclear potential for application.

Reviewer 2 Report

This paper proposes a novel expansion phenomenon of the oxidized liquid metal droplet, by applying different acoustic wave on the paper towel. The results can contribute to the triggering system with an applied acoustic wave using oxidized liquid metal.

However, the authors must implement some necessary improvements to the paper before it can be published. Please find the following comments to the authors.

  • In Fig. 1(b1~c3), it is better if you can give an scale for each picture.
  • In Fig. 3, it is better if you can add the experiment condition to titles.
  • The shape of liquid metal is symmetrical or not after expansion? How about the repeatability and symmetry of the shape of liquid metal after multi-repeatedly applying of acoustic sound waves? This is meaningful for application.
  • The time to maintain expansion in this paper should refer to the supporting effect of oxide layer on the expanded liquid metal with hollow structure. If the acoustic wave lasts longer, will it be easier to maintain the shape of the liquid metal if it can be more fully oxidized? If the temperature would influence the maintain effect?
  • There have existed some applications of Galinstan switch, but is not discussed in the introduction.(e.g. Liquid-metal capillary switch for electrical power application. Appl. Phys. Lett., 2020, 117: 263701)

Round 2

Reviewer 1 Report

submission acceptable for publication in its present form